# Traceless native chemical ligation of lipid-modified peptide surfactants by mixed micelle formation

Shuaijiang Jin[1], Roberto J. Brea[1], Andrew K. Rudd[1], Stuart P. Moon[2], Matthew R. Pratt [2] & Neal K. Devaraj [1✉]

Biology utilizes multiple strategies, including sequestration in lipid vesicles, to raise the rate and specificity of chemical reactions through increases in effective molarity of reactants. We show that micelle-assisted reaction can facilitate native chemical ligations (NCLs) between a peptide-thioester – in which the thioester leaving group contains a lipid-like alkyl chain – and a Cys-peptide modified by a lipid-like moiety. Hydrophobic lipid modification of each peptide segment promotes the formation of mixed micelles, bringing the reacting peptides into close proximity and increasing the reaction rate. The approach enables the rapid synthesis of polypeptides using low concentrations of reactants without the need for thiol catalysts. After NCL, the lipid moiety is removed to yield an unmodified ligation product. This micelle-based methodology facilitates the generation of natural peptides, like Magainin 2, and the derivatization of the protein Ubiquitin. Formation of mixed micelles from lipid-modified reactants shows promise for accelerating chemical reactions in a traceless manner.

[1] Department of Chemistry and Biochemistry, University of California, San Diego, 9500 Gilman Drive, La Jolla, CA 92093, USA. [2] Department of Chemistry, University of Southern California, Los Angeles, CA 90089, USA. ✉email: ndevaraj@ucsd.edu

Templated chemistry is a hallmark of biological processes, bringing reactants into close proximity with one another, thus greatly increasing effective molarity and dramatically raising the rate of reaction[1–3]. Many reactions that would otherwise not occur in practical time periods are facilitated by templating. Nucleic acid[4–8] and protein templates[9–11] are widely appreciated in biology and biomimetic chemistry, and a number of seminal studies have exploited such templates for programming the synthesis of complex molecules such as oligopeptides[12–14]. All living cells possess lipid membranes, which in addition to acting as compartments and barriers, also act as scaffolding structures for guiding biological reactions. By restricting substrates in two-dimensions, local concentration is increased. Signaling from clusters of membrane bound proteins benefits from this phenomenon, as does the biosynthesis of membrane components such as the bacterial cell wall. Given that templated protein synthesis is a principal tenet of the central dogma of molecular biology, it would be of particular interest to understand if lipid moiety mediated self-assembly of mixed micelles[15] can drive the selective formation of discreet polypeptides. Here, the mixed micelles would be composed of two different lipid detergents, bearing polar head groups that would be reactive with one another. If successful, mixed micelle assistance could provide a straightforward mechanism to accelerate sluggish chemical reactions and could be potentially be a useful synthetic method if mechanisms existed to make the process traceless through removal of lipid auxiliary groups.

To address these issues, we focused on applying lipid modification to accelerate oligopeptide coupling, specifically through native chemical ligation (NCL)[16]. NCL is a chemoselective reaction that occurs at neutral pH between two unprotected peptides, one bearing an N-terminal cysteine residue and the other containing a C-terminal thioester. NCL is a robust method[17–19] for the synthesis and derivatization of large peptides, proteins[20–24], and even nucleic acids[13,25,26] and phospholipids[27–29]. Despite its widespread use in chemical protein synthesis, NCL has several limitations. First, C-terminal peptide thioesters are commonly prepared as alkyl thioesters, which simplifies synthesis, purification and storage, but results in precursors with low reactivity[30]. Second, although some thiol-free NCL strategies have been developed recently[31–33], addition of thiol catalysts is often required in conventional NCL method to promote the in situ formation of more reactive thioesters[30,34]. Thiol catalysts, such as MPAA[30] introduced by Kent and coworkers, have been widely applied in synthetic protein chemistry[35,36]. Despite their proven robustness, thiol catalysts may have the potential for unwanted side reactions[31,33]. Furthermore, thiol additives can interfere with the NCL-desulfurization synthetic strategy[37,38] by acting as radical scavengers[39], which may necessitate a removal step prior to desulfurization[32,40]. Third, the modest reaction rates of NCL make it necessary to use millimolar concentrations of the substrate peptide fragments for practical application[13,34,41,42].

Development of a lipid-facilitated NCL methodology that allows the rapid production of ligated polypeptides, using low concentrations of reactants[43] and in the absence of thiol additive catalysts, would be of considerable value. Herein, we describe a strategy for lipid-facilitated acceleration of NCL via micelle mixing (Fig. 1). Our approach takes advantage of a peptide bearing a C-terminal alkyl thioester (1a–d, 5a,b or 6a,b), which is capable of forming micelles in aqueous solution. Addition of an Cys-peptide containing a photolabile lipid[44,45] moiety near the N-terminus (2a,b) leads to rapid NCL. We hypothesize that the close proximity[46–50] of the alkyl thioester and cysteine within formed micelles increases the rate of transthioesterification substantially, even at low concentrations of reagents (1 mM as standard concentration) and in the absence of thiol catalysts.

Although there are several reported cases[51–53] of facilitated NCL in hydrophobic environments or in lipid bilayers, in our work, the surfactants are the reacting peptide segments, to which lipid moieties are covalently attached. Micelles are formed by the lipid-modified reactants themselves, which upon addition to the same solution, fuse to form mixed micelles. During NCL, the lipid moiety attached to the thioester peptide is eliminated in the initial thiol-thioester exchange. After NCL, the lipid moiety maintained on the product can be removed by a photochemical process. Thus, a traceless synthetic system can be established. We demonstrate the utility of this strategy through the successful one-pot synthesis of the natural antimicrobial peptide Magainin 2 and derivatization of the natural protein Ubiquitin (Ubi). Our studies suggest that micelle-mixing assembly may be a broadly applicable method for the acceleration of chemical reactions.

## Results

**Design and synthesis of model peptide fragments.** To explore the potential of lipid-facilitated NCL, solid-phase peptide synthesis (SPPS) was used to construct the peptide LYRMG-OH, which was subsequently modified with linear aliphatic alkyl chains of different lengths to form the corresponding thioesters (Fig. 1)–C8 (1a), C16 (1b), and C18 (1c). We note that LYRMG peptide thioesters with Cn alkyl chain will henceforth be denoted as LYRMG-αCOSCn; e.g. LYRMG-αCOSC8 (1a). As a control, we prepared LYRMG-αCOSC2 (1d), a short-chain peptide thioester containing an ethyl moiety on its C-terminus, which lacks the necessary hydrophobicity for micelle formation. This behavior was confirmed by dynamic light scattering (DLS) measurements, showing that at 1 mM concentration, no micelle formation was observed for 1d. Additionally, we synthesized Cys-peptides (2a,b) using a standard Fmoc-SPPS protocol. These peptide segments were adapted from the peptide model sequence CRANK-OH[18]. In our case, we substituted the arginine for the unnatural amino acid Dap(PhCn), which bears a photolabile lipid group[44,45] at the $N_\beta$ position of 2,3-diaminopropionic acid (Dap). The photolabile group possesses a linear aliphatic alkyl chain of either C8 (2a) or C16 (2b). Peptide 2c, where the photolabile group is absent, was also prepared as a control.

**Kinetic analysis of lipid-facilitated model peptide ligations.** We tested the NCL reactions between the oligopeptide thioesters (1a–d) and the cysteine-functionalized peptide substrates (2a–c) in a thiol-additive-free aqueous phosphate buffered system at pH 7.0, containing tris(2-carboxyethyl) phosphine hydrochloride (TCEP·HCl) to maintain a reducing environment.

Cysteine-based oligopeptide CDap(PhC16)ANK (2b) was treated with various oligopeptide LYRMG-αCOSZ thioesters (1a–d) (Fig. 2a). The ligation of 1a and 2b provided product 3b in quantitative yield within 180 min. Half-maximal product formation was achieved in 10 min, and 95% ligation yield in 70 min. The reaction between 1b and 2b was slower, with quantitative yield within 960 min (16 h), but only 60% ligation yield in 180 min. The reaction of 1c and 2b was slow; less than 50% ligation yield could be achieved after 960 min. Use of thioester 1d, which lacks a long aliphatic chain, presented less than 20% ligation yield with 2b after 960 min, indicating the importance of mixed micelle self-assembly in accelerating the reaction.

Of the tested oligopeptide thioesters, 1a (C8) promoted the greatest acceleration of the NCL. Therefore, we studied its reactivity against various cysteine-based peptide analogs (2a–c) (Fig. 2b). The ligation rate of 2b and 1a, with ~60% ligation yield in 70 min and quantitative conversion after 420 min, was slower

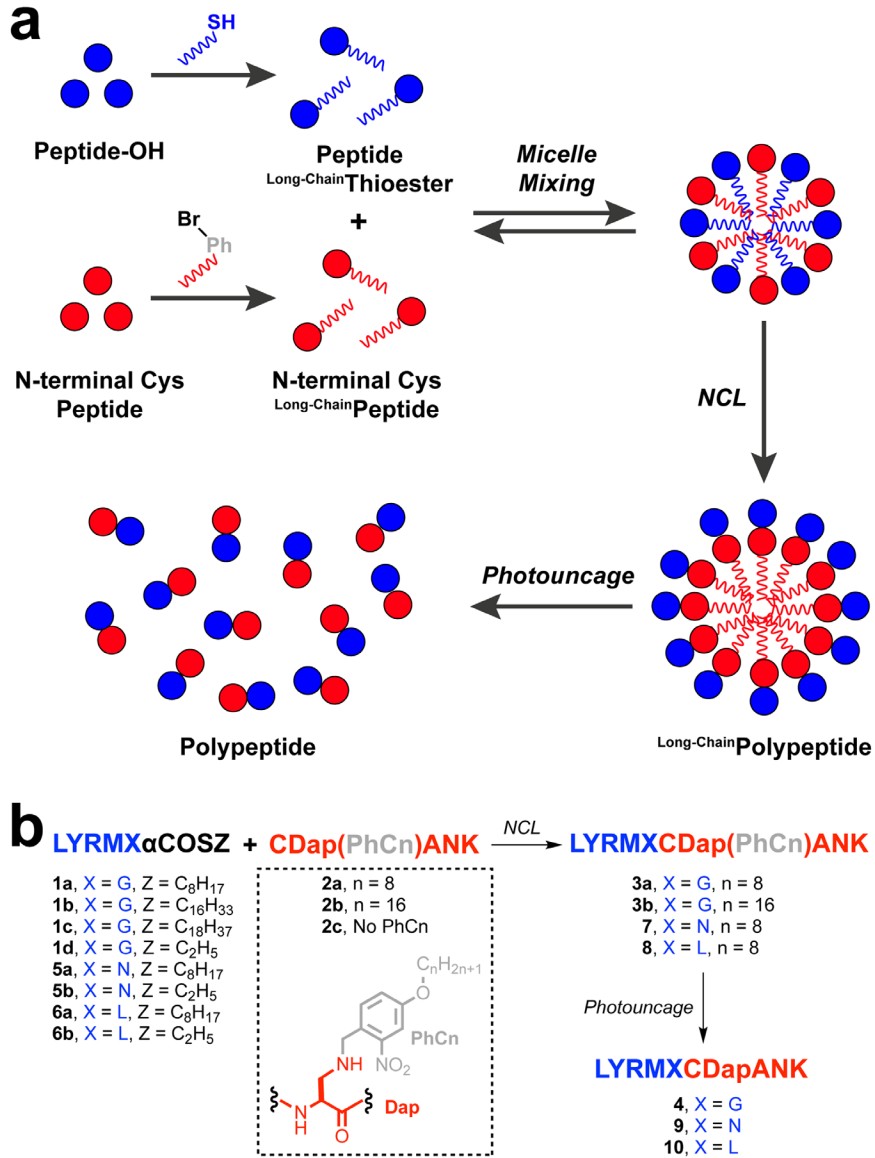

**Fig. 1 Traceless lipid-facilitated acceleration of NCL. a** Schematic representation of traceless lipid-facilitated NCL via micelle mixing. **b** NCL reaction between aliphatic alkyl peptide thioesters (**1a–c**, **5a** or **6a**) and photocaged cysteine-based peptides (**2a,b**), which yields the desired polypeptides (**3a,b**, **7** and **8**, respectively). Control reactions were performed with peptides containing short alkyl chains (**1d**, **2c**, **5b**, and **6b**). Subsequent photocleavage reaction of the ligated products (**3a,b**, **7** or **8**) leads to the formation of the uncaged polypeptides (**4**, **9**, and **10**, respectively).

when compared to **2a** and **1a**. As expected, the control reaction of **2c** with **1a** did not generate ligated product.

When keeping the Cys-peptide substrate constant (Fig. 2a), we observed that the ligation rate decreases with increasing alkyl thioester chain length [LYRMG-αCOSC18 (**1c**) < LYRMG-αCOSC16 (**1b**) < LYRMG-αCOSC8 (**1a**)]. Micelles composed of shorter alkyl thioester chains are predicted to have a higher rate of dissociation[54,55], resulting in facilitated micelle mixing, potentially explaining this result. However, when **1a** was reacted with various Cys-peptides differing in lipid chain length (Fig. 2b), we found that C16 reacted slightly faster than C8. At this time, the relationship between reaction rate and lipid chain length is unclear. Our system is complicated by the reaction between two different amphiphiles introduced as micelles, which form mixed micelles, as indicated by the observed reaction. Mixing two amphiphiles with different critical micelle concentrations (CMCs) can often lead to complex micelle formation behavior[56–59]. Although beyond the scope of the current work, future studies

using advanced characterization techniques could shed light on the kinetics of mixed micelle formation and their overall composition as a function of lipopeptide precursor.

To verify the proposed micelle formation-mixing mechanism, we carried out NCL experiments between **1a** and **2b** at multiple concentrations (Fig. 2c). When the concentrations of both peptide oligomers (1 mM, or 500 μM) were above their respective CMCs (~380 μM for **1a**, and ~130 μM for **2b**) (Supplementary Fig. 23), conditions under which mixed micelles consisting of both reactants are expected to form, ligations were fast (completed within 70 min). The curves for 1 mM and 500 μM were similar, which suggests that once the concentration is above the CMC, there is a limited increase in ligation rate with increasing reactant concentration, as one would expect if micelle formation is the critical parameter for achieving rate acceleration. When the concentration of reactants (250 μM) is above the CMC of **2b** but below the CMC of **1a** [conditions in which monomers of **1a** can fuse on micelles of **2b**], the ligation reaction proceeded

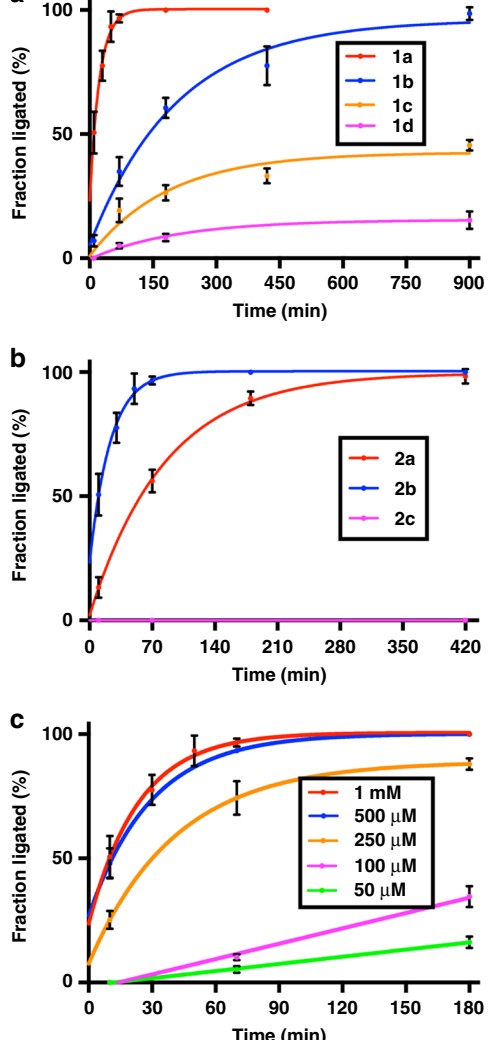

**Fig. 2 Kinetic measurements of NCL between LYRMG-Z (1a–d) and CDap (PhCn)ANK (2a–c). a** NCL reaction between **1a–d** (1 mM) and **2b** (1 mM). **b** NCL reaction between **1a** (1 mM) and **2a–c** (1 mM). **c** NCL reaction between **1a** (1 mM–50 μM) and **2b** (1 mM–50 μM). Peptides were ligated at room temperature and pH 7.0 in the presence of TCEP·HCl (10 mM). Decapeptide LYRMGCDap(PhCn)ANK (**3a,b**) formation was monitored over time using combined liquid chromatography (LC), mass spectrometry (MS), and evaporative light-scattering detection (ELSD) measurements. At each time point, the fraction ligated was determined by integration of the ligated product with detection at 210 nm as a fraction of the sum of {starting material cysteine peptide **2** + ligated product}. Error bars represent standard deviations (SD) ($n = 3$).

with an intermediate kinetic velocity (~70% ligation yield within 70 min). When the concentrations of the reactants (100 μM, or 50 μM) are below the CMCs of both reactants [conditions in which both **1a** and **2b** do not form micelles], ligations once again become slow (below 35% ligation yield for 100 μM, and below 17% ligation yield for 50 μM within 180 min). The significant difference in NCL kinetics at various reactant concentrations provides further evidence for the role of mixed micelle self-assembly in facilitating the reaction. In comparison, MPAA-catalyzed NCL reactions between the unlipidated precursors LYRMG-αCOSC2 (**1d**) and CDapANK (**2c**) were also performed (Supplementary Fig. 55). At 500 μM, the MPAA-catalyzed ligation was considerably slower than the lipid-facilitated ligation

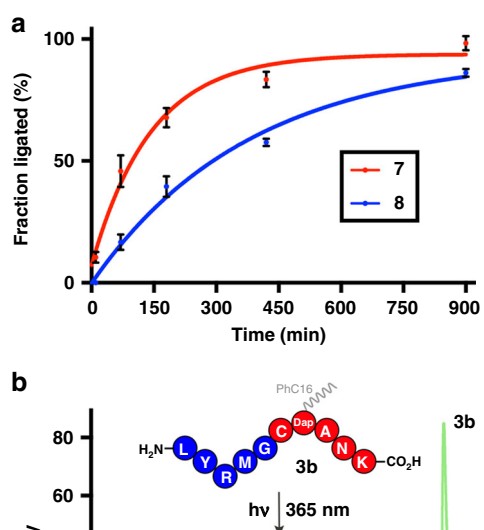

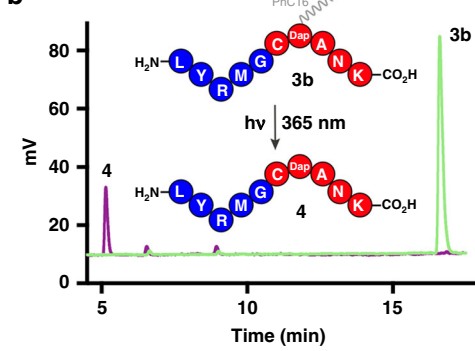

**Fig. 3 Various ligations (Asn-Cys, Leu-Cys) and photocleavage reaction. a** Kinetic measurements of NCL between the peptide thioesters LYRMN-αCOSC8 (**5a**) (1 mM) or LYRML-αCOSC8 (**6a**) (1 mM) and the peptide CDap(PhC16)ANK (**2b**) (1 mM). Decapeptide (**7** and **8**, respectively) formation was monitored over time using combined HPLC-ELSD-MS measurements. Error bars represent standard deviations (SD) ($n = 3$). **b** Photocleavage reaction of the auxiliary on NCL product LYRMGCDap (PhC16)ANK (**3b**) to generate the decapeptide LYRMGCDapANK (**4**). The photocleavage reaction was monitored using HPLC-ELSD-MS (raw ELSD data shown).

(Supplementary Fig. 36), which highlights the importance of mixed micelle self-assembly at low reactant concentrations.

**Scope of the lipid-facilitated peptide ligations.** To test the robustness and widen the applicability of our methodology, we generated more complex peptide ligation sites such as Asn-Cys and Leu-Cys (Fig. 3a). We synthesized peptide thioesters LYRMN-αCOSC8 (**5a**) and LYRML-αCOSC8 (**6a**) and treated them at room temperature with the N-terminal cysteine peptide CDap(PhC16)ANK (**2b**). The desired ligated polypeptides (**7** and **8**, respectively) were afforded in good yields (>80%) after 900 min. Control reactions of LYRMN-αCOSC2 (**5b**) and LYRML-αCOSC2 (**6b**) with **2b** did not generate the corresponding ligation products, supporting the role of lipid modification in acceleration of these ligations.

We next studied the effect of the distance between the lipid chains and the reactive centers. We prepared the peptide CADap (PhC16)NK (**S10**) (Supplementary Fig. 34), an analogue of **2b** where the unnatural amino acid containing the photolabile lipid group was located at the third position, increasing the distance between peptide reactive sites. As expected, kinetics experiments showed that NCL reaction between **1a** and **S10** was slower than the ligation with **2b**, demonstrating the importance of lipid chain positioning on reaction rate (Supplementary Figs. 35 and 54).

In classical NCL conditions, thiol catalysts, such as MPAA, are able to reverse nonproductive transthioesterification with the thiol moieties of non N-terminal cysteine residues[30]. NCL reaction between CK(PhC16)ANC (**S12**) (Supplementary Fig. 37)

and LYRMG-αCOSC8 (**1a**) demonstrated that our lipid-facilitated methodology is also compatible with the presence of non N-terminal cysteines (Supplementary Figs. 38, 39, and 56), possibly due to the lipid moiety directing reaction to the nearby N-terminal cysteine.

**Photoliberation of the lipid-free peptides**. Without purification, the nitrobenzyl photocaging group was removed using a standard photocleavage protocol[45], liberating the lipid-free peptide. This photochemical process allowed the generation of the uncaged peptide product **4** in nearly quantitative yield (Fig. 3b, Supplementary Figs. 17 and 51). Decapeptides **9** and **10** were obtained using analogous conditions (Supplementary Figs. 18 and 19, respectively). In this manner, a 'traceless' synthetic system can be established, in which both lipid modifications (the C-terminal thioester and the photocaging auxiliary group) on the starting materials (**1a**, **5a** or **6a**, and **2b**) are removed during the reaction process.

**Construction of native peptides**. Having achieved traceless peptide ligation by lipid-facilitated NCL, we sought to extend this approach to the synthesis of native peptides rather than Dap-containing products. By utilizing the amino acid Lys(PhCn), which bears the photolabile lipid group at the $N_\varepsilon$ position of the lysine, we successfully synthesized the decapeptide LYRMGCK-ANK (**S6**) (Supplementary Figs. 20–22 and 52).

**Lipid-facilitated total synthesis of Magainin 2**. Encouraged by these results, we next applied our methodology to the total synthesis of a natural peptide product. Magainin 2 is a 23aa cationic and amphipathic polypeptide that acts as a potent antibiotic in a variety of organisms (Fig. 4)[60,61]. This natural antimicrobial peptide has been reported to be a difficult sequence to synthesize by Fmoc chemistry SPPS[62,63]. We devised a straightforward one-pot synthesis of Magainin 2 based on our lipid-facilitated NCL methodology (Fig. 4). The peptide thioester GIGKFLHS-αCOSC8 (**11**) (Supplementary Fig. 27) was initially treated at 30 °C with the Cys-peptide CK(PhC16)KFGKAFV-GEIMNS (**12**) (Supplementary Figs. 26 and 53) to generate the ligated polypeptide **13** with 1.0 mM concentrations of precursors and in the absence of thiol catalysts (Supplementary Figs. 29–32). We note that sodium dodecyl sulfate (SDS) was required to completely solubilize the peptide fragment **12**. Addition of this anionic detergent provides a satisfying solution for the issues related with low solubility of certain lipopeptides. Additionally, a control reaction between GIGKFLHS-αCOSC2 (**S8**) (Supplementary Fig. 28) and peptide **12** under the same conditions generated no ligated product (Supplementary Fig. 33). By eliminating the need for thiol additives, our lipid-facilitated NCL strategy enables successive peptide ligation, photocleavage, and desulfurization in one-pot to afford the natural peptide Magainin 2 (**15**, 19% isolated yield for 3-step one-pot reaction).

**Lipid-facilitated expressed protein ligation of Ubiquitin**. Having demonstrated the utility of this approach in the synthesis of a natural peptide, we sought to apply our methodology to the C-terminal derivatization of proteins. Using expressed protein ligation, recombinant proteins bearing C-terminal thioesters can be generated and reacted with N-terminal cysteine peptides through NCL. Expressed protein ligation[64–67] has been extensively used for the site-specific introduction of probes, unnatural amino acids, and complex post-translational modifications. The recombinant protein-derived thioesters are usually prepared from a modified Cys-intein fusion protein, which can be cleaved with thiol to provide the corresponding thioester moiety. Thus, we

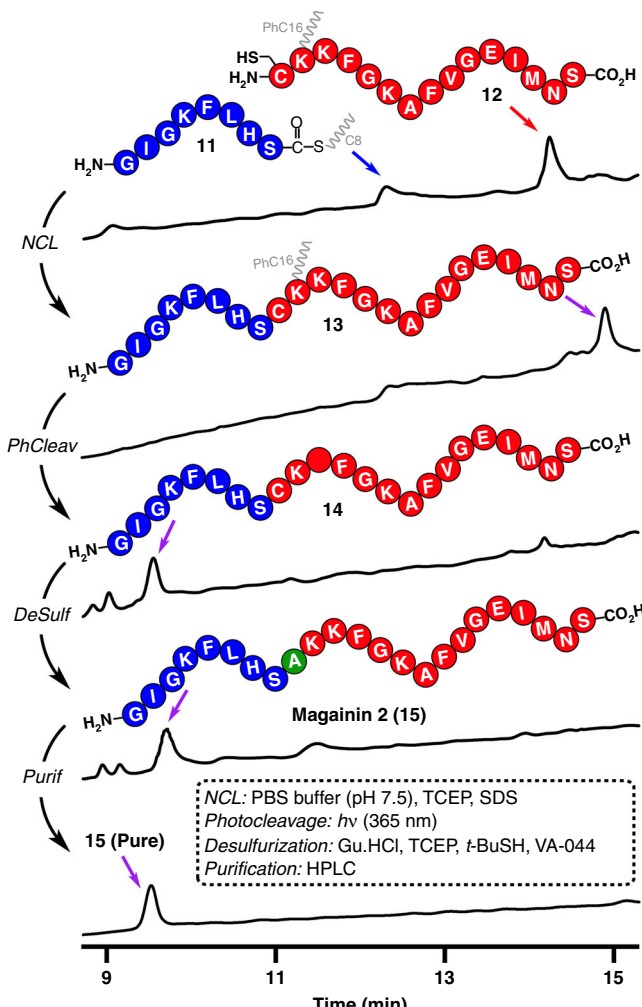

**Fig. 4 One-pot strategy for the total synthesis of Magainin 2 (15).** HPLC (210 nm) traces corresponding to the precursors, intermediates, and final product. Retention times were verified by MS.

obtained the Ubiquitin-derived thioester Ubi-αCOSC8 (**16**, Supplementary Fig. 58) through thiol cleavage from a recombinant Cys-intein fusion protein and transthioesterfication. The protein thioester Ubi-αCOSC8 (**16**, 0.5 mM) was subsequently treated with the Cys-peptide CK(PhC16)ANK (**S4**, 2 mM) to afford the ligation product, lipidated protein Ubi-CK(PhC16)ANK (**17**, 71% conversion), within 5 h at 37 °C without thiol additives (Fig. 5 and Supplementary Fig. 59). In contrast, the control reaction between Ubi-αCOSC8 (**16**) and peptide CKANK (**S13**, Supplementary Fig. 60) under the same conditions generated only 6% ligation product (Supplementary Figs. 61 and 62). These results demonstrate that our lipid-facilitated NCL methodology can be used in conjunction with expressed protein ligation to efficiently derivatize proteins. Indeed, our method could be particularly useful for the addition of lipid modifications onto full-length proteins.

**Discussion**
In summary, we have developed a mixed micelle-assisted NCL methodology that demonstrates the ability of lipid self-assembly to facilitate the synthesis of larger oligopeptides through accelerating NCL. This approach takes advantage of the presence of cleavable lipid chains in the reactants, leading to self-assembly of reactants into micelles, and acceleration of the ligation reaction by mixed micelle formation. Peptide ligation occurs quickly, at moderate concentration ranges, and in the absence of catalyst

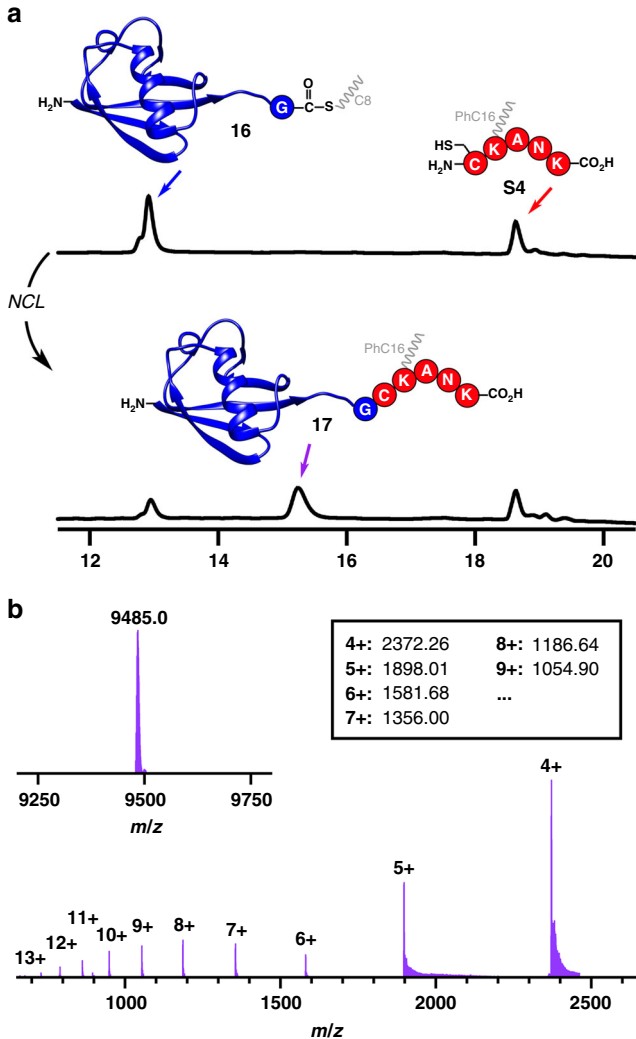

**Fig. 5 Derivatization of the natural protein Ubiquitin by lipid-facilitated NCL. a** HPLC (210 nm) traces corresponding to the precursors Ubi-αCOSC8 (**16**) and CK(PhC16)ANK (**S4**) (top) and the ligated product Ubi-CK(PhC16)ANK **17** (after 5 h of NCL reaction; bottom). Retention times were verified by MS. **b** ESI-TOF MS spectrum of the ligated product **17**. Deconvoluted spectrum calculated from this ESI-TOF spectrum is also shown in the top left. The deconvolution value (9485.0) corresponds with the molecular weight of the product.

additives. Moreover, ligation studies showed that a variety of amino acids were compatible with our methodology. The lipid-facilitated approach was successfully employed to synthesize the natural antimicrobial peptide Magainin 2 in one-pot. The hydrophobic effects conferred by the alkyl chains (C8 on the thioester segment, C16 on the Cys-peptide segment) were enough to efficiently facilitate the reactions with all the peptides studied. We believe that the hydrophobic character of the peptide fragments may also play a key role in the acceleration of the NCL. Therefore, incorporation of hydrophobic residues (especially close to the reaction centers) could considerably increase the rate of the peptide ligations[53]. Further studies using our approach will be focused on the construction of optimized hydrophobic lipopeptides that can drive fast NCL reactions. Since the cysteine-containing segments incorporate a relevant lipid modification, we foresee potential applications of our lipid-facilitated NCL strategy in the synthesis of structurally complex peptides, especially lipoproteins, which has been demonstrated by derivatization of

Ubiquitin. Furthermore, lipid-facilitated peptide ligation could be applied in the construction and selective modification of proteins and other biopolymers such as nucleic acids. More broadly, lipid-facilitated ligation could become a general method to accelerate reactions that would otherwise be too slow to be practical.

## Methods

**Synthesis of Fmoc-Dap(PhC16)-OH (S3a).** Here, we describe the representative procedure for the preparation of Cn-photocaged amino acids. Amino acid **S3a** was synthesized according to Supplementary Fig. 46.

4-(Hexadecyloxy)-1-methyl-2-nitrobenzene (**S1a**): a solution of 4-methyl-3-nitrophenol (5.00 g, 32.68 mmol) in $CH_3CN$ (200 mL) was treated with $K_2CO_3$ (8.13 g, 58.82 mmol) and 1-bromohexadecane (12.04 mL, 39.21 mmol). Then, the reaction mixture was stirred for 24 h at reflux (~82 °C; with a condenser). Afterwards, the resulting suspension was filtered to remove the $K_2CO_3$. The solvent was removed in vacuo and the crude was purified by flash chromatography (0%–10% EtOAc in hexanes) to afford **S1a** as a yellowish solid (11.34 g, 92%). [1]H NMR (500 MHz, $CDCl_3$, δ): 7.44 (d, J = 2.7 Hz, 1H), 7.16 (d, J = 8.4 Hz, 1H), 7.00 (dd, $J_1$ = 8.5 Hz, $J_2$ = 2.8 Hz, 1 H), 3.92 (t, J = 6.5 Hz, 2 H), 2.47 (s, 3H), 1.83-1.68 (m, 2 H), 1.43–1.36 (m, 2 H), 1.26-1.19 (m, 26 H), 0.83 (t, J = 6.8 Hz, 3H) (Supplementary Fig. 40). [13]C NMR (126 MHz, $CDCl_3$, δ): 157.61, 149.31, 133.37, 125.29, 120.44, 109.59, 68.63, 31.95, 29.74, 29.73, 29.72, 29.70, 29.69, 29.68, 29.63, 29.61, 29.58, 29.41, 29.40, 29.36, 29.07, 29.05, 25.96, 22.72, 19.80, 14.16 (Supplementary Fig. 40).

1-(Bromomethyl)-4-(hexadecyloxy)-2-nitrobenzene (**S2a**): a solution of **S1a** (2.00 g, 5.31 mmol) in $CCl_4$ (15 mL) placed on a flame-dried two-neck round bottom flask with a condenser was successively treated with recrystallized N-bromosuccinimide (NBS; 1.04 g, 5.84 mmol) and azobisisobutyronitrile (AIBN; 87.08 mg, 0.53 mmol). The resulting suspension was irradiated with white light (250 W Philips tungsten bulb) for 10 h. After this period of time, the light was removed and the reaction was cooled to rt. The obtained succinimide was filtered off and washed with $CH_2Cl_2$ (15 mL). The filtrate was then added to a separatory funnel and washed with $NaHCO_3$ (sat.) (10 mL), $H_2O$ (10 mL) and NaCl (sat.) (10 mL). The organic phase was dried over anhydrous $Na_2SO_4$ and concentrated in vacuo. The resulting crude material was then purified by flash chromatography (0-10% EtOAc in hexanes) to afford **S2a** as an off-white solid (1.45 g, 60%). [1]H NMR (500 MHz, $CDCl_3$, δ): 7.54 (d, J = 2.6 Hz, 1 H), 7.43 (d, J = 8.6 Hz, 1H), 7.11 (dd, $J_1$ = 8.5 Hz, $J_2$ = 2.7 Hz, 1H), 4.80 (s, 2 H), 4.01 (t, J = 6.5 Hz, 2H), 1.86-1.76 (m, 2 H), 1.51–1.42 (m, 2 H), 1.39–1.19 (m, 26 H), 0.88 (t, J = 6.8 Hz, 3 H) (Supplementary Fig. 41). [13]C NMR (126 MHz, $CDCl_3$, δ): 157.60, 149.31, 133.37, 125.29, 120.43, 109.60, 68.63, 31.80, 31.76, 29.31, 29.23, 29.04, 25.96, 22.67, 19.79, 14.12 (Supplementary Fig. 41).

(R)-2-((((9H-fluoren-9-yl)methoxy)carbonyl)amino)-3-((4-(hexadecyloxy)-2-nitro-benzyl)amino)propanoic acid (**S3a**): to a solution of **S2a** (1.00 g, 2.23 mmol) in MeOH:$CH_2Cl_2$ (4:1; 45 ml) was added $N_α$-Fmoc-L-2,3-diaminopropionic acid (Fmoc-Dap-OH; 873.68 mg, 2.68 mmol), and the mixture was stirred at rt for 5 min. The resulting suspension was then treated with N,N-diisopropylethylamine (DIEA; 1.17 mL, 6.69 mmol), which caused the suspension to gradually clear. The reaction mixture was stirred at rt for 20 h shielded from light. Afterwards, the solvent was removed in vacuo, while maintaining the temperature of the water bath around 20 °C to avoid potential decomposition and/or side reactions. Then, the obtained crude was purified by flash chromatography (0%–5% MeOH in $CH_2Cl_2$), affording **S3a** as a yellowish foam (1.01 g, 64%). [1]H NMR (500 MHz, $CDCl_3$, δ): 7.63 (d, J = 7.6 Hz, 2 H), 7.49 (dd, $J_1$ = 17.2 Hz, $J_2$ = 8.5 Hz, 3 H), 7.27 (td, $J_1$ = 7.4 Hz, $J_2$ = 4.1 Hz, 2 H), 7.19 (q, $J_1$ = 5.1 Hz, $J_2$ = 4.6 Hz, 3 H), 6.94 (d, J = 8.5 Hz, 1 H), 6.56 (br, 1 H), 4.31 (m, 3 H), 4.20 (t, J = 8.8 Hz, 1 H), 4.16-4.06 (m, 1 H), 4.06 (d, J = 7.5 Hz, 1 H), 3.73 (t, J = 6.5 Hz, 2 H), 3.50-3.34 (m, 2 H), 1.58 (p, J = 7.0 Hz, 2 H), 1.19 (d, J = 7.8 Hz, 28 H), 0.80 (t, J = 6.8 Hz, 3 H) (Supplementary Fig. 42). [13]C NMR (126 MHz, $CDCl_3$, δ): 172.87, 160.55, 156.51, 149.39, 143.81, 141.15, 135.39, 127.65, 127.11, 125.31, 120.46, 119.84, 117.51, 111.47, 68.91, 67.39, 52.06, 49.61, 48.84, 46.95, 31.95, 29.75, 29.60, 29.40, 28.91, 25.87, 22.75, 14.19 (Supplementary Fig. 42). HRMS (ESI-TOF, electrospray ionization time-of-flight mass analyzer) calculated for $[C_{41}H_{55}N_3O_7]^-$ ($[M-H]^-$) 700.3967, found 700.3963.

**Synthesis of CDap(PhC16)ANK (2b).** Here, we describe the representative procedure for the preparation of CDap(PhCn)ANK derivatives (General procedure I). Peptide **2b** was prepared manually by standard solid-phase peptide synthesis (SPPS) protocols (Supplementary Fig. 48). 2-Chlorotrityl chloride resin (500 mg; loading: 1.5 mmol/g) was soaked in anhydrous DCM (4 mL) for 30 min. The solvent was filtered off, and a solution of Fmoc-Lys(Boc)-OH (422 mg, 0.9 mmol) and DIEA (522 μL, 3 mmol) in anhydrous DCM (4 mL) was added to the resin. After 1 h, the solvent was filtered off and the resin was washed with DCM (4 mL). A mixture of DCM:MeOH:DIEA (8.5:1:0.5; 4 mL) was added and the resin was shaken for 30 min then washed with DCM (3 × 4 mL) and $Et_2O$ (4 mL). The resin was dried under high vacuum and the loading was determined by quantification of the Fmoc group. For this, a small portion of the resin (~2 mg) was treated with a solution of 20% piperidine in DMF (1 mL) for 30 min. To an aliquot of this solution (100 μL), 900 μL of DMF was added and the absorbance was read at 301 nm. The concentration of the dibenzofulvene-piperidine adduct was obtained

by using the extinction coefficients (ε) tabulated in the literature. Thus, the resin loading was estimated to be 0.66 mmol/g.

A portion of resin (0.1 mmol loaded Fmoc-Lys(Boc)-OH) was used for the synthesis of the desired peptide. The Fmoc group was removed by treatment with 20% piperidine in DMF (3 mL) for 30 min. The resin was washed with DMF (6 mL) and then treated with a solution of Fmoc-protected amino acid (0.4 mmol), $N,N,N',N'$-tetramethyl-$O$-(1$H$-benzotriazol-1-yl)uronium hexafluorophosphate (HBTU; 152.0 mg, 0.4 mmol), 1-hydroxybenzotriazole (HOBt; 69.0 mg, 0.4 mmol) and DIEA (139 μL, 0.8 mmol) in DMF (3 mL). The resin was shaken for 45 min and then washed with DMF (3 mL). The procedure was repeated with each corresponding amino acid [amino acids were introduced in sequence: Fmoc-Asn (Trt)-OH, Fmoc-Ala-OH, Fmoc-C$^{PhC16}$Dap-OH and Boc-Cys(Trt)-OH]. Note: When coupling with Fmoc-Dap(PhC16)-OH or Boc-Cys(Trt)-OH, the reaction time was prolonged to 6 h.

After the coupling of the last amino acid, the peptide was released from the resin and all of the protective groups were removed by treatment with freshly prepared cocktail K [trifluoroacetic acid (TFA):phenol:thioanisole:H$_2$O:1,2-ethanedithiol(EDT); 82.5: 5: 5: 5: 2.5; 3 mL per 100 mg of resin] for 2 h and then filtered. The resin was washed with TFA (0.5 mL) and the combined fractions were evaporated to 1–2 mL by bubbling argon. The concentrated solution was added dropwise to cold Et$_2$O (10 mL of Et$_2$O per ml of TFA). The resulting precipitate was centrifuged for 10 min at 1008 × g. The supernatant was discarded, then fresh Et$_2$O was added, and the suspension was sonicated and centrifuged. The resulting solid was dried under vacuum. The sample was dissolved in MeOH and purified by semipreparative HPLC using a C18 column [gradient of H$_2$O with 0.1% formic acid and MeOH with 0.1% formic acid 95:5 (0 min) to 5:95 (20 min)] to give CDap (PhC16)ANK (2b) as a white solid (8.98 mg, 10%). Analytical HPLC: $t_R$ = 2.85 min (5 to 95% Phase B over 4 min, then 95% Phase B for 7 min, Eclipse Plus C8 analytical column) (Supplementary Fig. 1). MS (ESI) [C$_{42}$H$_{73}$N$_9$O$_{10}$S] calculated: 896.5 [M + H]$^+$, 448.8 [M + 2H]$^{2+}$; found 896.4 [M + H]$^+$, 448.8 [M + 2H]$^{2+}$ (Supplementary Fig. 1).

**Synthesis of LYRMG-αCOSC8 (1a).** Here, we describe the representative procedure for the preparation of LYRMX-αCOSZ derivatives (General procedure II). Peptide **1a** was prepared manually by standard Fmoc chemistry solid phase peptide synthesis (SPPS) protocols (Supplementary Fig. 49). 2-Chlorotrityl chloride resin (500 mg; loading: 1.5 mmol/g) was soaked in anhydrous DCM (4 mL) for 30 min. The solvent was filtered off, and a solution of Fmoc-Gly-OH (268 mg, 0.9 mmol) and DIEA (522 μL, 3 mmol) in anhydrous DCM (4 mL) was added to the resin. After 1 h, the solvent was filtered off and the resin was washed with DCM (4 mL). A mixture of DCM:MeOH:DIEA (8.5:1:0.5; 4 mL) was added and the resin was shaken for 30 min, then washed with DCM (3 × 4 mL) and Et$_2$O (4 mL). The resin was dried under high vacuum and the loading was determined by quantification of the Fmoc group. For this, a small portion of the resin (~ 2 mg) was treated with a solution of 20% piperidine in DMF (1 mL) for 30 min. To an aliquot of this solution (100 μL) was added 900 μL of DMF and the absorbance was read at 301 nm. The concentration of the dibenzofulvene-piperidine adduct was obtained by using the extinction coefficients (ε) tabulated in the literature. Thus, the resin loading was estimated to be 0.70 mmol/g.

A portion of resin (0.1 mmol loaded Fmoc-Gly-OH) was used for the synthesis of the desired peptide. Fmoc group was removed by treatment with 20% piperidine in DMF (3 mL) for 30 min. The resin was washed with DMF (6 mL) and then treated with a solution of Fmoc-protected amino acid (0.4 mmol), HBTU (152.0 mg, 0.4 mmol), HOBt (69.0 mg, 0.4 mmol) and DIEA (139 μL, 0.8 mmol) in DMF (3 mL). The resin was shaken for 45 min and then washed with DMF (3 mL). The procedure was repeated with each corresponding amino acid [amino acids were introduced in sequence: Fmoc-Met-OH, Fmoc-Arg(Pbf)-OH, Fmoc-Tyr (tBu)-OH and Boc-Leu-OH].

After the coupling of the last amino acid, the protected peptide was released from the resin by treatment with freshly prepared mixture of 20% 1,1,1,3,3,3-hexafluoro-2-propanol (HFIP) in DCM (4 mL) for 1 h and then filtered. The resin was washed with DCM (2 × 1 mL) and the combined fractions were evaporated in vacuo, affording Boc-Leu-Tyr(tBu)-Arg(Pbf)-Met-Gly-OH as a white solid (58.6 mg, 56%). Analytical HPLC: $t_R$ = 2.9 min (5 to 95% Phase B over 1 min, then 95% Phase B for 9 min, Eclipse Plus C8 analytical column) (Supplementary Fig. 4). MS (ESI) [C$_{50}$H$_{78}$N$_8$O$_{12}$S$_2$] calculated 1047.5 [M + H]$^+$; found 1047.3 [M + H]$^+$ (Supplementary Fig. 4).

Boc-Leu-Tyr(tBu)-Arg(Pbf)-Met-Gly-OH (10.00 mg, 9.56 mmol) was dissolved in anhydrous DMF (5 mL) and then benzotriazole-1-yl-oxy-tris-pyrrolidinophosphonium hexafluorophosphate (PyBOP; 5.97 mg, 11.47 mmol) and DIEA (5.0 μL, 28.68 mmol) were successively added. After stirring for 5 min, 1-octanethiol (3.5 μL, 19.12 mmol) was added and the reaction mixture was stirred at rt for 24 h. Afterwards, the solvent was removed in vacuo. The resulting residue was dissolved in MeOH and purified by preparative HPLC giving Boc-Leu-Tyr (tBu)-Arg(Pbf)-Met-Gly-αCOSC8 as a white solid (9.64 mg, 88%).

Boc-Leu-Tyr(tBu)-Arg(Pbf)-Met-Gly-αCOSC8 (9.64 mg, 8.43 mmol) was deprotected by treatment with freshly prepared cocktail K (TFA:phenol:thioanisole: H$_2$O:EDT; 82.5: 5: 5: 5: 2.5; 0.5 mL). The solution was added dropwise to cold Et$_2$O (10 mL of Et$_2$O per ml of TFA). The resulting precipitate was centrifuged for 10 min at 1008 × g. The supernatant was discarded, then fresh Et$_2$O was added, and

the suspension was sonicated and centrifuged. The resulting solid was dried under vacuum. The sample was dissolved in MeOH and purified by semipreparative HPLC using a C18 column [gradient of H$_2$O with 0.1% formic acid and MeOH with 0.1% formic acid 95:5 (0 min) to 5:95 (20 min)] to give LYRMG-αCOSC8 (1a) as a white solid (4.20 mg, 68%). Analytical HPLC: $t_R$ = 4.35 min (5 to 95% Phase B over 8 min, Eclipse Plus C8 analytical column) (Supplementary Fig. 5). MS (ESI) [C$_{36}$H$_{62}$N$_8$O$_6$S$_2$] calculated 767.4 [M + H]$^+$, 384.2 [M + 2 H]$^{2+}$; found 768.3 [M + H]$^+$, 384.3 [M + 2 H]$^{2+}$ (Supplementary Fig. 5).

**Synthesis of LYRMGCDap(PhC16)ANK (3b).** Here, we describe the representative procedure for the preparation of LYRMGCDap(PhC16)ANK derivatives by NCL reaction (General procedure III). Peptide **3b** was synthesized according to Supplementary Fig. 50. The lyophilized model peptides **1a** (0.4 μmol, 1.0 equiv) and **2b** (0.4 μmol, 1.0 equiv) were dissolved in separate vials using freshly prepared ligation buffer [200 μL for each vial respectively. In all, 200 mM NaH$_2$PO$_4$ pH 7.0 containing 10 mM tris(2-carboxyethyl)phosphine (TCEP)]. After sonication of both mixtures for 10 minutes, the solutions from both vials were combined (final concentration of each peptide: 1 mM). The resulting solution was gently shaken at rt. Aliquots were taken at various time intervals and analyzed by analytical HPLC-MS. The yield was calculated based on the peak areas of [2b + 3b] versus the desired ligation product **3b** at 210 nm. Analytical HPLC: $t_R$ = 13.43 min (0% Phase B for 1 min, then 0 to 95% Phase B over 14 min, then 95% Phase B for 2 min, Eclipse Plus C8 analytical column) (Supplementary Fig. 13). MS (ESI) [C$_{70}$H$_{117}$N$_{17}$O$_{16}$S$_2$] calculated: 758.9 [M + 2 H]$^{2+}$, 506.3 [M + 3 H]$^{3+}$, 380.0 [M + 4 H]$^{4+}$; found 759.4 [M + 2 H]$^{2+}$, 506.5 [M + 3 H]$^{3+}$, 380.2 [M + 4 H]$^{4+}$ (Supplementary Fig. 13). The ligation product was purified by semipreparative HPLC using a C18 column [gradient of H$_2$O with 0.1% formic acid and MeOH with 0.1% formic acid 50:50 (0 min) to 5:95 (10 min to 15 min)], giving **3b** (0.5 mg, 81%) as a white solid.

**Synthesis of LYRMGCDapANK (4).** Here, we describe the representative procedure for the preparation of lipid-free peptides by photouncaging reaction (General procedure IV). Peptide **4** was synthesized according to Supplementary Fig. 51. When the native chemical ligation was completed, the reaction mixture containing the product LYRMGCDap(PhC16)ANK (3b) was directly transferred to a quartz tube. Then, the solution was irradiated with UV light (365 nm) for 5 min[45]. Afterwards, an aliquot was taken and analyzed by analytical HPLC-MS and the formation of the peptide **4** was observed. Analytical HPLC: $t_R$ = 5.0 min (0% Phase B for 1 min, then 0% to 95% Phase B over 19 min, then 95% Phase B for 2 min, Eclipse Plus C8 analytical column) (Fig. 3b). MS (ESI) [C$_{62}$H$_{101}$N$_{17}$O$_{16}$S$_2$] calculated: 571.3 [M + 2H]$^{2+}$, 381.2 [M + 3 H]$^{3+}$, 286.1 [M + 4 H]$^{4+}$; found 571.3 [M + 2 H]$^{2+}$, 381.4 [M + 3 H]$^{3+}$, 286.4 [M + 4 H]$^{4+}$ (Supplementary Fig. 17).

**Determination of critical micelle concentrations.** Each lipid-modified peptide (**1a**, **1b**, **2a**, **2b**) subjected to CMC determination was initially lyophilized, and then dissolved in separate vials by addition of freshly prepared ligation buffer. The resulting aqueous solutions (100 μL) of LYRMG-αCOSC8 (**1a**, 5 μM, 25 μM, 100 μM, 250 μM, 500 μM, 750 μM, 1 mM, 1.5 mM, 2 mM) and CDap(PhC16)ANK (**2b**, 10 μM, 25 μM, 50 μM, 100 μM, 150 μM, 250 μM, 375 μM, 500 μM, 750 μM) were analyzed by DLS in order to determine the CMCs[68]. This technique allowed us to measure the CMCs at 25 °C, since the scattered light intensity (measured at 90°) is dependent on the molecular weight and size of the particle, it increases when monomers start to aggregate in solution. Therefore, in our particular case, light scattering intensity values were plotted versus the concentration of the lipid-modified peptide acting as a surfactant (Supplementary Figs. 23–25). Light-scattering intensity was very low and constant when the concentration was low, denoting the presence of monomers in solution. After sequential increases in concentration, light-scattering intensity increased, and the CMC was determined by intersecting the two extrapolated lines corresponding to monomer and micelle regions.

**Total synthesis of Magainin 2 (15).** Magainin 2 was synthesized on a one-pot-3-step reaction (NCL → photocleavage→desulfurization) from peptides GIGKFLHS-αCOSC8 (**11**) and CK(PhC16)KFGKAFVGEIMNS (**12**).

Preparation of CK(PhC16)KFGKAFVGEIMNS (**12**): Peptide **12** was synthesized according to Supplementary Fig. 53. Using the general procedure I, the fully protected peptidyl-resin Boc-Cys(Trt)-Lys(Alloc)-Lys(Boc)-Phe-Gly-Lys (Boc)-Ala-Phe-Val-Gly-Glu(tBu)-Ile-Met-Asn(Trt)-Ser(tBu)-2-Chlorotrityl chloride resin (**S7a**) was obtained (loading: 0.46 mmol/g). Then, the Alloc group on the peptidyl-resin (**S7a**, 50.0 mg) was removed with tetrakis(triphenylphosphine) palladium(0) [Pd(PPh$_3$)$_4$; 0.2 equiv] in the presence of phenylsilane [PhSiH$_3$; 20 equiv]. Once the deprotection was completed (3 h; 2 times), the resin was washed with sodium diethyldithiocarbamate trihydrate in DMF (0.02 M, 15 min; three times) to remove palladium residues. Afterwards, the resin **S7b** was suspended in DMF, and 2 equiv of DIEA and 1 equiv of 1-(bromomethyl)-4-(hexadecyloxy)-2-nitrobenzene (**S2a**) were successively added. After shaking at rt for 8 h, the resin was successively washed with DMF and DCM, and finally dried under vacuum for 12 h.

The peptide was next unprotected and released from the resin (**S7c**) by treatment with freshly prepared cocktail K (TFA:phenol:thioanisole:H$_2$O:EDT; 82.5: 5: 5: 5: 2.5; 3 mL per 100 mg of resin) for 2 h, and then filtered. The resin was washed with TFA (0.5 mL) and the combined fractions were evaporated to 1–2 mL by bubbling argon. The concentrated solution was added dropwise to cold Et$_2$O (10 mL of Et$_2$O per ml of TFA). The resulting precipitate was centrifuged for 10 min at 1008 x $g$. The supernatant was discarded, then fresh Et$_2$O was added, and the suspension was sonicated and centrifuged. The resulting solid was dried under vacuum. The sample was dissolved in MeOH and purified by HPLC using a Zorbax SB-C18 semipreparative column [50% to 95% Phase B for 10 min, then 95% Phase B for 8 min], obtaining CK(PhC16)KFGKAFVGEIMNS (**12**) as a white solid (4.10 mg, 9%) [Note: to solubilize and transfer **12**, an MeCN:H$_2$O (1:1, with 1% TFA) solution was used]. Analytical HPLC: $t_R$ = 2.58 min (50 to 95% Phase B over 1 min, then 95% Phase B for 5 min, Eclipse Plus C8 analytical column) (Supplementary Fig. 26). MS (ESI) [C$_{97}$H$_{156}$N$_{20}$O$_{23}$S$_2$] calculated: 1017.55 [M + 2 H]$^{2+}$, 678.7 [M + 3 H]$^{3+}$; found 1018.5 [M + 2 H]$^{2+}$, 678.8 [M + 3 H]$^{3+}$ (Supplementary Fig. 26).

Preparation of GIGKFLHS-αCOSC8 (**11**): using the general procedure II, 1.31 mg of the 8-mer thioester peptide GIGKFLHS-αCOSC8 (**11**) were obtained [white solid, 18% for two steps from 10.00 mg of Boc-Gly-Ile-Gly-Lys(Boc)-Phe-Leu-His(Trt)-Ser (tBu)-OH]. Analytical HPLC: $t_R$ = 2.31 min (50 to 95% Phase B over 1 min, then 95% Phase B for 5 min, Eclipse Plus C8 analytical column) (Supplementary Fig. 27). MS (ESI) [C$_{48}$H$_{79}$N$_{11}$O$_9$S] calculated: 986.6 [M + H]$^+$, 493.8 [M + 2 H]$^{2+}$, 329.5 [M + 3 H]$^{3+}$; found 986.5 [M + H]$^+$, 494.0 [M + 2 H]$^{2+}$, 329.6 [M + 3 H]$^{3+}$ (Supplementary Fig. 27).

One-pot-3-step reaction (NCL → photocleavage → desulfurization): Native chemical ligation (NCL): The lyophilized peptides GIGKFLHS-αCOSC8 (**11**, 0.4 μmol) and CK(PhC16)KFGKAFVGEIMNS (**12**, 0.4 μmol) were dissolved in separate vials (200 μL for each vial respectively) using freshly prepared and degassed ligation buffer (200 mM NaH$_2$PO$_4$ containing 10 mM TCEP + 2 mM SDS, then adjusted to pH 7.5). After sonication of both mixtures for 10 min, the solutions were combined (final concentration of each peptide: 1 mM). The resulting solution was gently shaken at 30 °C for 24 h under argon. An aliquot was taken and analyzed by HPLC-MS (Fig. 4). GIGKFLHSCK(PhC16) KFGKAFVGEIMNS (**13**) was verified to be the peak with $t_R$ = 14.94 min (5% Phase B for 1 min, then 5 to 95% Phase B over 14 min, and then 95% Phase B for 2 min, Eclipse Plus C8 analytical column). MS (ESI) [C$_{137}$H$_{217}$N$_{31}$O$_{32}$S$_2$] calculated: 958.5 [M + 3 H]$^{3+}$, 965.9 [M + 2 H + Na]$^{3+}$, 719.2 [M + 4 H]$^{4+}$; found 958.2 [M + 3 H]$^{3+}$, 966.0 [M + 2 H + Na]$^{3+}$, 719.3 [M + 4 H]$^{4+}$ (Supplementary Fig. 29).

Photocleavage: Without purification, the previous NCL reaction mixture was directly transferred to a quartz tube. Then, the photocleavage reaction was performed following the general procedure IV. An aliquot was taken and analyzed by HPLC-MS (Fig. 4). GIGKFLHSCKKFGKAFVGEIMNS (**14**) was verified to be the peak with $t_R$ = 9.66 min (5% Phase B for 1 min, then 5% to 95% Phase B over 14 min, and then 95% Phase B for 2 min, Eclipse Plus C8 analytical column). MS (ESI) [C$_{114}$H$_{180}$N$_{30}$O$_{29}$S$_2$] calculated: 1249.7 [M + 2 H]$^{2+}$, 833.4 [M + 3 H]$^{3+}$, 625.3 [M + 4 H]$^{4+}$; found 1249.6 [M + 2 H]$^{2+}$, 833.6 [M + 3 H]$^{3+}$, 625.2 [M + 4 H]$^{4+}$ (Supplementary Fig. 30).

Desulfurization: After photocleavage, to the vial containing 400 μL of the reaction mixture was added 40 μL of 0.5 M Bond-breaker® TCEP solution (neutral pH), 40 μL of 6 M guanidinium chloride (GuHCl) aqueous solution, 8 μL of 2-methyl-2-propanethiol and 8 μL of radical initiator 2,2′-azobis[2-(2-imidazolin-2-yl)propane] dihydrochloride (VA-044; 0.1 M in degassed water). Then, the reaction mixture was stirred at 37 °C. After 120 min, an aliquot was taken and analyzed by HPLC-MS (Fig. 4), confirming that the desulfurization was completed by MS. The desired product (main peak) was purified and verified to be GIGKFLHSAKKFGKAFVGEIMNS (Magainin 2, **15**). Analytical HPLC: $t_R$ = 9.58 min (5% Phase B for 1 min, then 5 to 95% Phase B over 14 min, and then 95% Phase B for 2 min, Eclipse Plus C8 analytical column). MS (ESI) [C$_{114}$H$_{180}$N$_{30}$O$_{29}$S] calculated: 822.8 [M + 3 H]$^{3+}$, 617.3 [M + 4 H]$^{4+}$; found 822.9 [M + 3 H]$^{3+}$, 617.3 [M + 4 H]$^{4+}$ (Supplementary Figs. 31). The final product was purified by semipreparative HPLC using a C18 column [gradient of H$_2$O with 0.1% formic acid and MeOH with 0.1% formic acid 95:5 (0 min) to 5:95 (15 min to 18 min)], giving **15** (0.2 mg, 19% for 3-step one-pot reaction) as white solid. Note: In conventional conditions[69,70], SDS [usually 1% (w/v), 35 mM] is difficult to be removed by HPLC. In our case, the concentration of SDS is 2 mM, which is far below the SDS concentration used in conventional methods. Exhaustive analysis of the obtained HPLC-MS traces showed that the retention time of SDS is 15.5 min, and the final product was SDS free.

The expected mass of the purified product (Magainin 2, **15**) was also corroborated by high resolution mass spectrometry (ESI-TOF) analysis (Supplementary Fig. 32).

The full spectra of the synthesis of Magainin 2 (**15**) (lipid-facilitated NCL → photocleavage→desulfurization→purification) are shown in Supplementary Fig. 57.

**Derivatization of Ubiquitin.** Ubi-CK(phC16)ANK (**17**) was synthesized by NCL between the protein thioester Ubi-αCOSC8 (**16**) and CK(PhC16)ANK (**S4**).

Preparation of Ubi-αCOSC8 (**16**): BL21 (DE3) cells were transformed with the Ubi-AvaE-6xHis plasmid by 30 s heat shock at 42 °C and plated overnight on an ampicillin LB-agar plate (100 μg/mL) at 37 °C. Overnight LB cultures (100 μg/mL ampicillin) were inoculated with a single colony from the plate and grown overnight at 37 °C with shaking at 7 × $g$. Growth TB cultures (100 μg/mL ampicillin) were inoculated with a small amount of overnight culture (1 mL LB/100 mL TB) and incubated at 37 °C with shaking at 7 × $g$ until mid-log phase was reached (OD$_{600}$ = 0.6–0.8) prior to induction by the addition of 0.5 mM IPTG. Cells were incubated at 25 °C with 7 × $g$ shaking overnight. Bacteria was then harvested by centrifugation at 6000 × $g$, 4 °C for 15 min. Pellets were resuspended in lysis buffer (50 mM Na$_2$HPO$_4$, 300 mM NaCl, 5 mM imidazole, 2 mM PMSF, 2 mM TCEP·HCl, pH 7.4) and probe-sonicated (70% amplitude, 30 s on/30 s off, 12 min total, 4 °C). The soluble fraction was separated by centrifugation at 7000 × $g$ for 1 h at 4 °C and incubated with pre-equilibrated Co-NTA affinity resin for 1 h at 4 °C with rocking. Beads were rinsed thoroughly with wash buffer (50 mM Na$_2$HPO$_4$, 300 mM NaCl, 20 mM imidazole, 2 mM TCEP·HCl, pH 7.4) before elution of target protein with elution buffer (50 mM Na$_2$HPO$_4$, 300 mM NaCl, 250 mM imidazole, 2 mM TCEP·HCl, pH 7.4). The eluate was concentrated, and buffer exchanged into 1X DPBS via Amicon Ultra 15 mL centrifugal filters (3 K MWCO) and subjected to sodium 2-mercaptoethanesulfonate (MesNa) thiolysis (250 mM MesNa, 2 mM TCEP·HCl, pH 7.2) for 48 h at 25 °C. The ubiquitin-thioester was then separated from the intein by C4 reverse-phase purification using a Biotage Isolera Spektra FLASH system. Characterization of the ubiquitin-thioester was performed using C4 analytical RP-HPLC and ESI-MS. Lyophilized Ubi-MesNa was dissolved to 5 mM in DMSO containing 50 molar equivalents of 1-octanethiol and 2 mM TCEP·HCl. The pH was adjusted to 7.0 and incubated for 1 h at 25 °C with monitoring by C4 analytical RP-HPLC. A separate product peak appeared with a later retention time in an ~75% yield. Unreacted Ubi-MesNa could be purified and recycled in subsequent thiol exchange reactions to improve yield. The Ubi-αCOSC8 was separated from the MesNa thioester by C4 reverse-phase purification using a Biotage Isolera Spektra FLASH system. Prior to lyophilization, the Ubi-αCOSC8 was purified by semipreparative HPLC using a C18 column [20% MeCN (0.1% TFA) for 1 min, then 20%–95% MeCN (0.1% TFA) over 15 min] to give Ubi-αCOSC8 (**16**). Analytical HPLC: $t_R$ = 12.30 min [5% MeCN (0.1% TFA) for 1 min, then 5%–70% MeCN (0.1% TFA) over 19 min, Eclipse Plus C8 analytical column]. The expected mass of Ubi-αCOSC8 (**16**) was corroborated by mass spectrometry (ESI-TOF) (Supplementary Fig. 58).

Native chemical ligation to prepare Ubi-CK(PhC16)ANK (**17**): The lyophilized peptides Ubi-αCOSC8 (**16**, 10 nmol) and CK(PhC16)ANK (**S4**, 40 nmol) were dissolved in separate vials (10 μL for each vial respectively) using freshly prepared and degassed ligation buffer (200 mM NaH$_2$PO$_4$ containing 20 mM TCEP + 10 mM SDS, then adjusted to pH 7.5). The solutions were combined (Fig. 5a, top). Then, both vials were carefully washed with the combined solution by pipetting three times. The resulting solution was gently shaken at 37 °C for 5 h under argon. An aliquot was taken and analyzed by HPLC (Fig. 5a, bottom). The HPLC elution of peak with $t_R$ = 15.30 min [5% MeCN (0.1% TFA) for 1 min, then 5%–70% MeCN (0.1% TFA) over 19 min; Eclipse Plus C8 analytical column] was collected, and then subjected to analysis by mass spectrometry (ESI-TOF) (Fig. 5 and full spectra shown in Supplementary Fig. 59). The expected mass of Ubi-CK(PhC16) ANK (**17**) was corroborated (Fig. 5b).

## Data availability
The authors declare that the data supporting the findings of this study are available within the paper and its Supplementary Information files, and, also available from the corresponding author upon reasonable request.

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

## Acknowledgements

This material is based upon work supported by the National Science Foundation (EF-1935372) and Human Frontier Science Program (HFSP) research grant RGY0066/2017. Roberto J. Brea thanks the Human Frontier Science Program (HFSP) for his Cross-Disciplinary Fellowship. Matthew R. Pratt thanks his National Institutes of Health Grant (R01GM114537).

## Author contributions

S.J., R.J.B, A.K.R., and N.K.D. conceived and designed the experiments. S.J., R.J.B, and S.P.M. performed the experiments. S.J., R.J.B, A.K.R., M.R.P., and N.K.D. analyzed the data. S.J., R.J.B, A.K.R., and N.K.D. co-wrote the paper.

## Competing interests

The authors declare no competing interests.
