## [Peer Review File · Nature Communications]

Reviewers' comments:

Reviewer #1 (Remarks to the Author):

The manuscript 'Traceless Lipid-Templated Ligation of Peptides' by Jin, et al., reports an intriguing approach to facilitating the native chemical ligation (NCL) of a peptide-thioester with a Cys-peptide. The two reacting peptide segments are modified with lipid-like long alkyl chain moieties so that the 'lipid'-modified peptides act as micelle forming surfactants. Mixing the separate micelles leads to mixed micelle formation and facilitated NCL reaction in the absence of thiol catalyst. The ligated product contains only the lipid moiety from the Cys-peptide; the other lipid moiety was lost with the thioester leaving group in initial thiol-thioester exchange. Use of a photolabile linker provides for facile removal of that sole lipid, providing for an overall 'traceless' NCL reaction.

Because of the novelty of this ingenious approach to facilitating NCL and its potential usefulness for chemical protein synthesis, I strongly recommend publication in Nature Communications.

The experimental data reported justify (most of) the conclusions drawn by the authors. However, the title of the paper, the abstract, and the text of the manuscript require significant revisions in order to more clearly and explicitly describe the concepts underlying this novel approach to NCL, and the 'mixed micelle' mechanism proposed for the reaction.

Title: the concept described in the paper is simply "Traceless Native Chemical Ligation of Lipid-Modified Peptide Surfactants by Mixed Micelle Formation". The title should reflect that. As it stands, use of the term 'template' is at best unclear, and was confusing to this reader.

Abstract: The opening statement that lipid-facilitated reactions are less explored is not correct. There is a considerable body of literature on the use of micelle-facilitated synthetic organic chemistry. For a recent review, see:

Paprocki D, Madej A, Koszelewski D, Brodzka A and Ostaszewski R (2018) 'Multicomponent Reactions Accelerated by Aqueous Micelles.'

Front. Chem. 6:502.

doi: 10.3389/fchem.2018.00502

A suggested reworking of the Abstract to highlight the essential aspects of the concept described in the paper and its importance is attached as a pdf file.

Text: The ideas in the paper are not conveyed in a logical sequence. As currently written, a reader has to mine clues from diverse parts of the text and even the Supporting Information in order to form an intellectual construct comprising the concept being reported. The concept should be clearly stated up front: traceless (i.e. gives an unmodified product) native chemical ligation of lipid-modified peptide surfactants, by formation of mixed micelles within which the NCL reaction is thought to occur, followed by photolytic removal of the remaining lipid moiety. The text should then be laid out as is schematically illustrated in (modified) Figure 1(a): 1. preparation of the peptide-thioester reactant, lipid modified in the thiol leaving group moiety; and, preparation of the lipid-modified Cys-peptide reactant, with a photocleavable link (between the lipid moiety and the peptide) to enable removal of the sole lipid moiety that remains after NCL; 3. formation of mixed micelles, the NCL reaction, followed by photolytic removal of the residual lipid moiety to give the unmodified ligation product.

In the Introductory paragraphs, there is no reference to and discussion of prior reports of NCL carried out in lipid bilayers, by Hunter & Kochemdoerfer (Native Chemical Ligation of Hydrophobic Peptides in Lipid Bilayer Systems, Bioconjugate Chem., Vol. 15, No. 3, 2004) and separately by Otaka, et al. (Facile synthesis of membrane-embedded peptides utilizing lipid bilayer-assisted chemical ligation. Chem Commun (Camb). 2004 Aug 7;(15):1722-3, in which it is stated:

"Chemical ligation under lipid bilayer-assisted conditions proceeded efficiently, in spite of the fact that the apparent concentration of the segments is 20 times lower (0.05 mmol dm³) than under other conventional conditions (1 mmol dm³). This result can probably be attributed to a "concentration effect" of sparingly soluble TMD peptides into the lipid bilayer. ").

Nor is the extensive literature of surfactant micelle-assisted organic synthesis cited and discussed (for a review, see: Paprocki D, Madej A, Koszelewski D, Brodzka A and Ostaszewski R (2018) Multicomponent Reactions Accelerated by Aqueous Micelles. Front. Chem. 6:502. doi: 10.3389/fchem.2018.00502)

Note that the concept explored in the current work is different from both of these lipid/surfactant approaches - in the work reported in the current manuscript, the 'surfactants' are the reacting peptide segments themselves, to which 'lipid' moieties are covalently attached. Micelles are formed by the lipid-modified reactants themselves, which upon addition to the same solution, (presumably) fuse to form 'mixed micelles' [Note: for the readers not familiar with this terms, define 'mixed micelle': e.g. ". . . the term mixed micelle means a micelle composed of surfactants capable themselves of forming micelles"] within which the NCL reaction occurs. It is important to state this clearly and unambiguously in the text.

Other Important Points:

** The statement elaborated in the first paragraph on page 7 with respect selectivity for an N-terminal Cys residue in the presence of a second Cys in the same peptide segment, as far as I can tell is not supported by data provided: as described in Supplementary Fig. 56, compound S11 was prepared by standard NCL in 6M Gu.HCl with MPAA as thiol catalyst, NOT by micelle-assisted reaction.**

*How does the addition of SDS affect micelle formation for GIGKFLHS-C2? Would adding the same concentration of SDS to the reaction with GIGKFLHS-C8 affect that reaction (i.e. a truly comparable 'Control')?

*Use a distinct font, and italics, to distinguish the thioester leaving group from the amino acid single letter code - and explain it, the first time it is used. That is, rather than 'LYRMN-C8' use something like 'LYRMN-SC8', or even better 'LYRMN-αCOSC8'

*Having the DhCn superscripted before Dap is not consistent with IUPAC nomenclature & conventional usage for modified amino acids (it makes it look like the N-terminal Cys is side chain modified): use '-Dap(PhCn)-'

An annotated pdf file with comments and questions from an initial reading of the manuscript is attached. A suggested revision of the Abstract is attached as a pdf file.

Signed: Stephen BH Kent

Reviewer #1 Additional Comments

ARTICLE:

1. 'While numerous studies have utilized peptide and nucleic acid templates to accelerate reactions, synthesis dependent on lipid-based templates has been less explored.' - This is not a correct statement. There is a body of literature on the use of micelle accelerated/facilitated synthetic oorganic chemistry. See: Paprocki D, Madej A, Koszelewski D, Brodzka A and Ostaszewski R (2018) Multicomponent Reactions Accelerated by Aqueous Micelles. *Front. Chem.* 6:502. doi: 10.3389/fchem.2018.00502
2. 'Hydrophobic lipid modifications promote the formation of dynamic mixed micelles' Why 'dynamic'? Evidence?
3. 'All living cells possess lipid membranes, which in addition to acting as compartments and barriers, also act as scaffolding structures for templating biological reactions' - 'templating' not precisely the correct term; maybe 'constraining', 'proteoining', 'guiding', ?
4. 'In contrast to nucleic acid and protein templates, there have been far less explorations in the use of lipid functional groups to template chemistry and most previous studies have been limited to the air-water interface or lead to non-specific product formation' - not clear how this allusion to the central dogma of molecular biology t is linked to the understanding of micelle mediated ligation of peptides. Explicate or remove.
5. 'Development of a lipid-templated NCL methodology that allows the rapid production of ligated polypeptides, using low concentrations of reactants and in the absence of thiol additive catalysts, would be of considerable value.' - 'template' is not the right word? <https://www.merriam-webster.com/dictionary/template>
Definition of template. 1a(1) : a gauge, pattern, or mold (such as a thin plate or board) used as a guide to the form of a piece being made. (2) : a molecule (as of DNA) that serves as a pattern for the generation of another macromolecule (such as messenger RNA)
6. 'As a control, we prepared LYRMG-C2' - italicize and use a distinct font (e.g. smaller font) to distinguish from the one letter amino acid code
7. 'Our system is complicated by the reaction between two different amphiphiles, which form mixed micelles' - define 'mixed micelle'; is it a micelle containing both reactants that were introduced as separate micelles. Evidence for this happening?
8. 'NCL reaction between CPhC16 KANC (S12) (Supplementary Fig. 37) and LYRMG-C8 (1a) demonstrated that our lipid-templated methodology is also compatible with the presence of non N-terminal cysteines (Supplementary Figs. 38, 39 and 56), possibly due to the lipid template directing reaction to the nearby N-terminal cysteine.' - No. As described in Supplementary Fig. 56, compound S11 was prepared by standard NCL in 6M Gu.HCl with MPAA as thiol catalyst.
9. 'Addition of this anionic detergent provides a satisfying solution for the issues related with low solubility of certain lipopeptides.' - How does the addition of SDS affect micelle formation? Would adding the same concentration of SDS to the reaction with GIGKFLHS-C8 affect that reaction (i.e. a truly comparable 'Control')?
10. 'Boc-Leu-Tyr(tBu)-Arg(Pbf)-Met-Gly-C8' - IUPAC: the dash represents a peptide bond

11. 'After sequential increases in concentration, light-scattering intensity increased, and the CMC was determined by intersecting the two straight lines corresponding to monomer and micelle regions.'- the region after CMC is NOT a straight line
12. 'The desired product (main peak) was purified and corroborated to be GIGKFLHSAKKFGKAFVGEIMNS (Magainin 2, 15)'- No. It had a mass consistent with being that product. You did not confirm the sequence of the product. Or did you, by MS-MS?
- 13.

Add a diagram at the top of this Figure 2(a) showing schematically what the blue dot-squiggle and red dot-squiggle chemical structures are [cf. Fig.4]. Then have an arrow labeled 'separate micelle formation' , etc.

14. 'Control reactions were performed with peptides containing short alkyl chains (1d, 2c, 5b and 6b)'- SDS needed? Apparently, so not true 'Controls' under identical conditions.
15. 'Decapeptide (7 and 8, respectively) formation was monitored over time using combined HPLC-ELSD-MS measurements'- define 'ELSD'

SUPPLEMENTARY INFORMATION:

1. 'As shown in Supplementary Fig. 56, CPhC16KANC (S10) and LYRMG-C8 (1a) were subjected to the general procedure III for NCL, and LYRMGCPhC16KANC (S11) was synthesized.'- No. As described in Supplementary Fig. 56, compound S11 was prepared by standard NCL in 6M Gu.HCl with MPAA as thiol catalyst.
- 2.

Supplementary Figure 38. Kinetic measurement of NCL between LYRMG-C8 (**1a**, 1 mM) and C¹⁶KANC (**S12**, 1mM).

Analytical HPLC of reaction at t=0 and at t=75 min needed, to show what the MS data derive from.

Reviewer #2 (Remarks to the Author):

In this study, the authors utilize removable side-chain lipid anchors on two reactive peptides to enhance the efficiency of native chemical ligation by taking advantage of micelle formation. In contrast to other templated approaches, the authors incorporate photocleavable lipid chains on peptides, which lead to a rate enhancement for NCL. The authors test this concept first in model peptides and later transfer it to the synthesis of an antimicrobial peptide.

In general, I regard the idea of using removable hydrophobic anchors to accelerate reaction rates in ligation reactions interesting; however, I am not able to recommend publication of this work in its current stage, due to several factors.

- The introduction of this paper is mainly written in the context of templated reactions, for sure an important aspect, but the authors fail to cite important work, in which NCL is performed in hydrophobic environments or even by insertion of peptide motifs in liposomes or lipid bilayers (to name a few: J. Pept. Sci. 2010, 16, 558; Bioconjugate Chem. 2004, 15, 437; Bioconjugate Chem. 2007, 18, 590). Furthermore, the literature references to the current limitations of NCL are outdated (page 3 top) since several protocols appeared for thiol additive-free ligation reactions.

- I think that the study itself would benefit from a more thorough investigation of ligation rates at more challenging ligation sites. So far, the authors have only tried three C-terminal amino acids in the thioester peptide and already observed significantly reduced ligation rates when compared to C-terminal Gly. Furthermore, it seems that lipid modification is mandatory at the residue following the N-terminal Cysteine, which further limits synthetic utility. Overall, I am not convinced that this ligation concept adds many benefits to the ligation repertoire by the presented data.

- Closely related, how do the peptide solubility properties influence the ligation rate? The authors apply the synthesis to Magainin, which is known to be polar and potentially amphiphilic itself. In order to be useful for the community other challenging sequences with different solubility properties, synthetically interesting building blocks (PTMs?) should be tested.

- I noted that (isolated) yields are only mentioned in the SI. These should be added to the main text. Also the author describe yields of the alkyl-peptides seem to be rather low. Was there a problem with purification? How do the yields compare to standard C2 synthesis?

- Are side-products observed after photocleavage? Looking at Figure 4 it is hard to tell. Spectra spanning the whole HPLC run should be included in the SI.

In summary, I find this study interesting and well performed. A nice proof of concept study is presented to utilize removable lipid substituents in to accelerate native chemical ligation.

Nevertheless, in order to be published in a high impact paper like Nature Comm I would expect a more convincing application on its synthetic utility and/or broad applicability. Therefore, I think that publication in a more specialized journal (Bioconj Chem, OBC) is more suitable at this stage.

Reviewer #3 (Remarks to the Author):

The paper by N. Devaraj and coworker entitled "Traceless Lipid-Templated Ligation of Peptides" present a smart idea for enhancing the molar concentration of lipidated peptides, one with lipid chain on the Cys-peptide, and the second with lipid C-thioester peptide and as a result enhance the NCL rate at somewhat lower concentration. I like the idea and it makes sense. However, I think the manuscript needs revisions and clarifications before it can be accepted for publication.

General:

Templated NCL should be cited especially the work of Ghadiri and others.

words such like "very" and "extremely" should be avoided.

please add the "PhCn" after the amino acid Dap not before and make it superscript like this:

Dap(PhCn).

Another important issue is regarding the purification of the lipidated peptides by HPLC. To me this is problematic most of the time, as the lipidated peptides tend to stick to the column irreversibly. Sometimes hydrophobic peptides are hard to handle and purify, so the lipidated forms will cause

problems most of the time, unless I am missing something. The authors should show the utility of this method for the synthesis of a protein rather than a short peptide Magainin 2, a 23-residue long polypeptide.

Specific:

in the introduction "even at low concentration of reagents" please add the units, mM or μ M here. the word "Discovered" when speaking about MPAA should be replaced with reported.

The inhibition of desulfurization reaction by aryl thiol was first suggested by Dawson (2010 - work on deselenization)

The explanation that C8 (hot spot - not too long not too short) gave the best performance is acceptable.

Page 5: the ligation rate of 2a and 1a was slower ..., with ~60% ligation yield in 70 min. The figure shows rather ~90% not 60%.

NCL reactions are normally performed at 1-3 mM concentrations. So 1 mM used in NCL in this report should be considered the standard conc. not low mM conc.

Payne and coworkers recently reported the NCL reactions at nM concentrations. This paper should be cited.

Responses to the Reviewer's Comments:

Reviewer 1

The manuscript 'Traceless Lipid-Templated Ligation of Peptides' by Jin, et al., reports an intriguing approach to facilitating the native chemical ligation (NCL) of a peptide-thioester with a Cys-peptide. The two reacting peptide segments are modified with lipid-like long alkyl chain moieties so that the 'lipid'-modified peptides act as micelle forming surfactants. Mixing the separate micelles leads to mixed micelle formation and facilitated NCL reaction in the absence of thiol catalyst. The ligated product contains only the lipid moiety from the Cys-peptide; the other lipid moiety was lost with the thioester leaving group in initial thiol-thioester exchange. Use of a photolabile linker provides for facile removal of that sole lipid, providing for an overall 'traceless' NCL reaction.

Because of the novelty of this ingenious approach to facilitating NCL and its potential usefulness for chemical protein synthesis, I strongly recommend publication in Nature Communications.

The experimental data reported justify (most of) the conclusions drawn by the authors. However, the title of the paper, the abstract, and the text of the manuscript require significant revisions in order to more clearly and explicitly describe the concepts underlying this novel approach to NCL, and the 'mixed micelle' mechanism proposed for the reaction.

Title: the concept described in the paper is simply "Traceless Native Chemical Ligation of Lipid-Modified Peptide Surfactants by Mixed Micelle Formation". The title should reflect that. As it stands, use of the term 'template' is at best unclear, and was confusing to this reader.

Response 1 *We thank the reviewer for the extremely helpful comments and kind suggestions. We have removed the term 'template' from the manuscript. Also, at the reviewer's suggestion, we have changed the title from 'Traceless Lipid-Templated Ligation of Peptides' to 'Traceless NCL of Lipid-Modified Peptide Surfactants by Mixed Micelle Formation'.*

Abstract: The opening statement that lipid-facilitated reactions are less explored is not correct. There is a considerable body of literature on the use of micelle-facilitated synthetic organic chemistry. For a recent review, see:

Paprocki D, Madej A, Koszelewski D, Brodzka A and Ostaszewski R (2018) 'Multicomponent Reactions Accelerated by Aqueous Micelles.'

Front. Chem. 6:502.

doi: 10.3389/fchem.2018.00502

A suggested reworking of the Abstract to highlight the essential aspects of the concept described in the paper and its importance is attached as a pdf file.

Response 2: *We thank the reviewer for the comments and thoughtful reworking of the Abstract. We have cited the review paper by Paprocki et al. regarding micelle-facilitated synthetic organic chemistry in the new version of manuscript and have revised the Abstract based on the reviewer's revisions. Please note, due to the abstract word limit (150 words), we had to make some alterations to the suggested abstract the reviewer was kind enough to provide. Nonetheless, we believe that we have faithfully preserved the key suggested edits.*

Text: The ideas in the paper are not conveyed in a logical sequence. As currently written, a reader has to mine clues from diverse parts of the text and even the Supporting Information in order to form an intellectual construct comprising the concept being reported. The concept should be clearly stated up front: traceless (i.e. gives an unmodified product) native chemical ligation of lipid-modified peptide surfactants, by formation of mixed micelles within which the NCL reaction is thought to occur, followed by photolytic removal of the remaining lipid moiety. The text should then be laid out as is schematically illustrated in (modified) Figure 1(a): 1. preparation of the peptide-thioester reactant, lipid modified in the thiol leaving group moiety; and, preparation of the lipid-modified Cys-peptide reactant, with a photocleavable link (between the lipid moiety and the peptide) to enable removal of the sole lipid moiety that remains after NCL; 3. formation of mixed micelles, the NCL reaction, followed by photolytic removal of the residual lip moiety to give the unmodified ligation product.

Response 3: *We apologize for the confusion and thank the reviewer for the comments. We have revised the Introduction to improve reporting of our concept. We have revised Figure 1(a) based on reviewer's suggestion.*

In the Introductory paragraphs, there is no reference to and discussion of prior reports of NCL carried out in lipid bilayers, by Hunter & Kochemdoerfer (Native Chemical Ligation of Hydrophobic Peptides in Lipid Bilayer Systems, Bioconjugate Chem., Vol. 15, No. 3, 2004) and separately by Otaka, et al. (Facile synthesis of membrane-embedded peptides utilizing lipid bilayer-assisted chemical ligation. Chem Commun (Camb). 2004

Aug 7;(15):1722-3, in which it is stated: “Chemical ligation under lipid bilayer-assisted conditions proceeded efficiently, in spite of the fact that the apparent concentration of the segments is 20 times lower (0.05 mmol dm³) than under other conventional conditions (1 mmol dm³). This result can probably be attributed to a “concentration effect” of sparingly soluble TMD peptides into the lipid bilayer. “).

Nor is the extensive literature of surfactant micelle-assisted organic synthesis cited and discussed (for a review , see: Paprocki D, Madej A, Koszelewski D, Brodzka A and Ostaszewski R (2018)

Multicomponent Reactions Accelerated by Aqueous Micelles.

Front. Chem. 6:502.

doi: 10.3389/fchem.2018.00502)

Note that the concept explored in the current work is different from both of these lipid/surfactant approaches - in the work reported in the current manuscript, the ‘surfactants’ are the reacting peptide segments themselves, to which 'lipid' moieties are covalently attached. Micelles are formed by the lipid-modified reactants themselves, which upon addition to the same solution, (presumably) fuse to form 'mixed micelles' [Note: for the readers not familiar with this terms, define ‘mixed micelle’: e.g. “. . . the term mixed micelle means a micelle composed of surfactants capable themselves of forming micelles”] within which the NCL reaction occurs. It is important to state this clearly and unambiguously in the text.

Response 4: *We thank the reviewer for the comments. We have cited the above-mentioned studies of facilitated NCL in hydrophobic environments (or in lipid bilayers) and the review paper by Paprocki et al. regarding micelle-facilitated synthetic organic chemistry in our revised manuscript. We also thank the reviewer for pointing out the novelty of our work compared to previous studies. In the revised manuscript, we now highlight the difference between our work and previous studies based on the reviewer’s suggestion. We have also taken care to define the meaning of mixed micelles and that micelles are formed by the lipid-modified reactants themselves.*

Other Important Points:

** The statement elaborated in the first paragraph on page 7 with respect selectivity for an N-terminal Cys residue in the presence of a second Cys in the same peptide segment, as far as I can tell is not supported by data provided: as described in Supplementary Fig. 56, compound S11 was prepared by standard NCL in 6M Gu.HCl with MPAA as thiol catalyst, NOT by micelle-assisted reaction.**

Response 5: *We thank the reviewer for identifying this oversight. In Supplementary Fig. 56, the compound LYRMGCK(PhC16)ANC (S11) was prepared without thiol additive. We have corrected this error in the revised Supplementary Information.*

*How does the addition of SDS affect micelle formation for GIGKFLHS-C2? Would adding the same concentration of SDS to the reaction with GIGKFLHS-C8 affect that reaction (i.e. a truly comparable 'Control')?

Response 6: *We thank the reviewer for their comments. As to the control reaction, we did use the ligation buffer with 2 mM SDS (the same condition for the NCL in the synthesis of Magainin 2) to maintain comparable reaction conditions. We have modified the text and ordering of sentences to make this point less ambiguous. In the new version of the Supplementary Information, we describe the conditions for the control reaction as: 'The lyophilized peptides GIGKFLHS- α COSC2 (S8, 1 mM) and CK(PhC16)KFGKAFVGEIMNS (12, 1 mM) were subjected to the same NCL condition (same ligation buffer with 2 mM SDS and temperature) for the synthesis of Magainin 2 in Methods.', to avoid misunderstanding.*

*Use a distinct font, and italics, to distinguish the thioester leaving group from the amino acid single letter code - and explain it, the first time it is used. That is, rather than 'LYRMN-C8' use something like 'LYRMN-SC8', or even better 'LYRMN- α COSC8'

Response 7: *We thank the reviewer for the comments. In the new version of manuscript, we denote the thioester leaving group in peptide thioester as '- α COSCn' based on the reviewer's suggestion. For instance, 'LYRMN-C8' was changed to 'LYRMN- α COSC8' based on reviewer's suggestion.*

*Having the DhCn superscripted before Dap is not consistent with IUPAC nomenclature & conventional usage for modified amino acids (it makes it look like the N-terminal Cys is side chain modified): use '-Dap(PhCn)-'

Response 8: *We thank the reviewer for the comments. Based on reviewer's suggestion, in the new version of manuscript, all of the lipidated Dap or lysine are denoted as 'Dap(PhCn)' or 'K(PhCn)'.*

An annotated pdf file with comments and questions from an initial reading of the manuscript is attached. A suggested revision of the Abstract is attached as a pdf file.

Signed: Stephen BH Kent

Response 9: *We would like to thank Prof. Kent for his detailed revision and comments on the manuscript and Supplementary Information, and kind rework of the Abstract. The authors greatly appreciate the time Prof. Kent spent reviewing our paper. We have revised the manuscript and Supplementary Information based on the annotations in the attached PDF files and have revised the Abstract based on Prof. Kent's reworking.*

Reviewer 2

In this study, the authors utilize removable side-chain lipid anchors on two reactive peptides to enhance the efficiency of native chemical ligation by taking advantage of micelle formation. In contrast to other templated approaches, the authors incorporate photocleavable lipid chains on peptides, which lead to a rate enhancement for NCL. The authors test this concept first in model peptides and later transfer it to the synthesis of an antimicrobial peptide.

In general, I regard the idea of using removable hydrophobic anchors to accelerate reaction rates in ligation reactions interesting; however, I am not able to recommend publication of this work in its current stage, due to several factors.

- The introduction of this paper is mainly written in the context of templated reactions, for sure an important aspect, but the authors fail to cite important work, in which NCL is performed in hydrophobic environments or even by insertion of peptide motifs in liposomes or lipid bilayers (to name a few: J. Pept. Sci. 2010, 16, 558; Bioconjugate Chem. 2004, 15, 437; Bioconjugate Chem. 2007, 18, 590). Furthermore, the literature references to the current limitations of NCL are outdated (page 3 top) since several protocols appeared for thiol additive-free ligation reactions.

Response 1: *We thank the reviewer for the comments. We have cited the above-mentioned studies of facilitated NCL in hydrophobic environments or in lipid bilayers in our revised manuscript. Also, we would like to point out the novelty of our work compared to previous studies. In our work, the 'surfactants' are the reacting peptide*

segments, to which lipid moieties are covalently attached. Micelles are formed by the lipid-modified reactants themselves, which upon addition to the same solution, form 'mixed micelles'. In the revised manuscript, we mentioned the difference between our work and previous studies. Furthermore, we cited several works regarding the recently developed methods of thiol additive-free NCL.

- I think that the study itself would benefit from a more thorough investigation of ligation rates at more challenging ligation sites. So far, the authors have only tried three C-terminal amino acids in the thioester peptide and already observed significantly reduced ligation rates when compared to C-terminal Gly. Furthermore, it seems that lipid modification is mandatory at the residue following the N-terminal Cysteine, which further limits synthetic utility. Overall, I am not convinced that this ligation concept adds many benefits to the ligation repertoire by the presented data.

Response 2: We thank the reviewer for the comments. We have demonstrated our methodology works at ligation sites with C-terminal -Gly, -Asn, -Leu and -Ser (synthesis of Magainin 2). The reduced ligation rates are due to the hinderance of the C-terminal amino acids, which is in accordance with a previous report [Proc. Natl. Acad. Sci. U.S.A. **96**, 10068-10073 (1999)]. In response to the reviewer's comment 'lipid modification is mandatory at the residue following the N-terminal Cysteine', we would respectfully point out that we synthesized the pentapeptide CADap(PhC16)NK (**S10**), an analogue of CDap(PhC16)ANK (**2b**) where the unnatural amino acid containing the photolabile lipid group Dap(PhC16) was located at the third position (not the residue following the N-terminal Cysteine), and proved that it also afforded the desired ligation product using our lipid-facilitated methodology, albeit at slightly lower rates (Supplementary fig. 35). Overall, our work is an initial fundamental study of the use of lipids to promote micelle based peptide ligation, where the lipopeptides both form the micelle and are reactants. We feel the impact of this work is in the novelty of the approach. In our study, the reacting peptide segments act as the 'surfactants', to which lipid moieties are covalently attached. Micelles are formed by the lipid-modified reactants themselves, which upon addition to the same solution, fuse to form 'mixed micelles'. More broadly speaking, we foresee our methodology could become a general method to accelerate reactions that would otherwise be too slow to be practical.

- Closely related, how to the peptide solubility properties influence the ligation rate? The authors apply the synthesis to Magainin, which is known to be polar and potentially amphiphilic itself. In order to be useful for the community other challenging sequences with different solubility properties, synthetically interesting building blocks (PTMs?) should be tested.

Response 3: We thank the reviewer for the comments. While it is hard to predict the influence of the peptide solubility properties on the ligation rate, in our study we found that issues related to poorly soluble peptide precursors could be resolved by using the detergent SDS. In the synthesis of Magainin 2 and derivatization of Ubiquitin, SDS was applied to solubilize precursors that are otherwise insoluble in the aqueous ligation buffer. Both reactions proceeded smoothly with addition of SDS.

With regards to trying other challenging substrates, in the revised version of our manuscript, we have added an exciting new experiment demonstrating the derivatization of a natural protein Ubiquitin (in collaboration with the Matthew Pratt Lab at USC, who are now coauthors). The protein thioester Ubi- α COSC8 (**16**, 0.5 mM), derived from recombinant Cys-intein fusion protein, was treated with the Cys-peptide CK(PhC16)ANK (**S4**, 2 mM) to afford the ligation product, lipidated protein Ubi-CK(PhC16)ANK (**17**, 71% conversion), within 5 hours at 37°C without thiol additives (Fig. 5 and Supplementary Fig. 59). In contrast, the control reaction between Ubi- α COSC8 (**16**) and peptide CKANK (**S13**, Supplementary Fig. 60) under the same conditions generated only 6% product (Supplementary Fig. 61 and 62). These results demonstrate the robustness of our lipid facilitated NCL methodology by using it in conjunction with expressed protein ligation to efficiently derivatize proteins. Indeed, our method could be particularly useful for the addition of lipid modifications onto full-length proteins.

- I noted that (isolated) yields are only mentioned in the SI. These should be added to the main text. Also the author describe yields of the alkyl-peptides seem to be rather low. Was there a problem with purification? How do the yields compare to standard C2 synthesis?

Response 4: We thank the reviewer for the comments. In the revised manuscript, we added the isolated yield of Magainin 2 in the main text. Also, as mentioned, the synthetic yields of -C8, -C16 and -C18 peptide thioesters are relatively low compared to the yield of -C2 peptide thioester. This is due to the lower efficiency of the thioester forming step (from C-terminal -OH peptide to C-terminal thioester peptide), which is possibly caused by the bulkiness of long-chain alkyl thiol.

- Are side-products observed after photocleavage? Looking at Figure 4 it is hard to tell. Spectra spanning the whole HPLC run should be included in the SI.

Response 5: We thank the reviewer for the comments. In the new version of Supplementary Information, the full HPLC spectra (Supplementary Fig. 57) related to Figure 4 (synthesis of Magainin 2) are included.

In summary, I find this study interesting and well performed. A nice proof of concept study is presented to utilize removable lipid substituents in to accelerate native chemical ligation. Nevertheless, in order to be published in a high impact paper like Nature Comm I would expect a more convincing application on its synthetic utility and/or broad applicability. Therefore, I think that publication in a more specialized journal (Bioconj Chem, OBC) is more suitable at this stage.

Reviewer 3

The paper by N. Devaraj and coworker entitled "Traceless Lipid-Templated Ligation of Peptides" present a smart idea for enhancing the molar concentration of lipidated peptides, one with lipid chain on the Cys-peptide, and the second with lipid C-thioester peptide and as a result enhance the NCL rate at somewhat lower concentration. I like the idea and it makes sense. However, I think the manuscript needs revisions and clarifications before it can be accepted for publication.

General:

Templated NCL should be cited especially the work of Ghadiri and others.

Response 1: *We thank the reviewer for the comments. We have cited the papers of templated NCL by Ghadiri in the manuscript (Ref. 9 and 11).*

words such like "very" and "extremely" should be avoided.

Response 2: *We thank the reviewer for the comments. We have removed words like "very" and "extremely" in the new version of manuscript.*

please add the "PhCn" after the amino acid Dap not before and make it superscript like this: Dap(PhCn).

Response 3: *We thank the reviewer for the comments. In the new version of manuscript, all of the lipidated Dap or lysine are denoted as 'Dap(PhCn)' or 'K(PhCn)'.*

Another important issue is regarding the purification of the lipidated peptides by HPLC. To me this is problematic most of the time, as the lipidated peptides tend to stick to the column irreversibly. Sometimes hydrophobic peptides are hard to handle and purify, so the lipidated forms will cause problems most of the time, unless I am

missing something. The authors should show the utility of this method for the synthesis of a protein rather than a short peptide Magainin 2, a 23-residue long polypeptide.

Response 4: *We thank the reviewer for the comments. We did not experience any difficulties with the purification of the lipidated peptides studied in this paper. Even highly hydrophobic peptides and protein, such as CK(PhC16)KFGKAFVGEIMNS (12) and Ubi- α COSC8 (16)], were successfully handled and purified using the conditions reported in the Supplementary Information.*

To demonstrate the utility of this method for the synthesis of a protein, in our revised manuscript, we detail a new experiment showing the derivatization of the natural protein Ubiquitin (in collaboration with the Matthew Pratt Lab at USC, who are now coauthors). The protein thioester Ubi- α COSC8 (16, 0.5 mM), derived from recombinant Cys-intein fusion protein, was treated with the Cys-peptide CK(PhC16)ANK (S4, 2 mM) to afford the ligation product, lipidated protein Ubi-CK(PhC16)ANK (17, 71% conversion), within 5 hours at 37°C without thiol additives (Fig. 5 and Supplementary Fig. 59). In contrast, the control reaction between Ubi- α COSC8 (16) and peptide CKANK (S13, Supplementary Fig. 60) under the same conditions generated only 6% ligated product (Supplementary Fig. 61 and 62). These results demonstrate the robustness of our lipid facilitated NCL methodology, by using it in conjunction with expressed protein ligation to efficiently derivatize proteins. Indeed, our method could be particularly useful for the addition of lipid modifications onto full-length proteins. We believe this new experiment raises the impact of the study, and we thank the reviewer for suggesting this experiment.

Specific:

in the introduction "even at low concentration of reagents" please add the units, mM or μ M here.

Response 5: *We thank the reviewer for the comments. In the introduction, we have now noted '1 mM as standard concentration' in our methodology.*

the word "Discovered" when speaking about MPAA should be replaced with reported.

Response 6: *We thank the reviewer for the comments. We replaced the word "Discovered" with "reported".*

The inhibition of desulfurization reaction by aryl thiol was first suggested by Dawson (2010 - work on deselenization)

Response 7: *We thank the reviewer for the comments. We have cited Prof. Dawson's elegant paper 'Traceless ligation of cysteine peptides using selective deselenization. Angew. Chem. Int. Ed. 49, 7049-7053 (2010)' in the new version of manuscript.*

The explanation that C8 (hot spot - not too long not too short) gave the best performance is acceptable.

Page 5: the ligation rate of 2a and 1a was slower, with ~60% ligation yield in 70 min. The figure shows rather ~90% not 60%.

Response 8: *We thank the reviewer for bringing this to our attention. We have corrected this oversight in our revised manuscript.*

NCL reactions are normally performed at 1-3 mM concentrations. So 1 mM used in NCL in this report should be considered the standard conc. not low mM conc.

Response 9: *We thank the reviewer for the comments. In the new version of our manuscript, we state '1 mM as the standard concentration' in our methodology to avoid misunderstanding.*

Payne and coworkers recently reported the NCL reactions at nM concentrations. This paper should be cited.

Response 10: *We thank the reviewer for the comments. We have cited Prof. Payne's paper 'Peptide Ligation at High Dilution via Reductive Diselenide-Selenoester Ligation. J. Am. Chem. Soc. 142, 1090-1100 (2020)' in the new version of our manuscript.*

Responses to the Editorial Comments:

In order to better summarize the described research, we have reworded the title of the manuscript from "Traceless Lipid-Templated Ligation of Peptides" to "Traceless NCL of Lipid-Modified Peptide Surfactants by Mixed Micelle Formation".

REVIEWERS' COMMENTS:

Reviewer #1 (Remarks to the Author):

The authors have appropriately addressed all the concerns and suggestions in my original report.

Stephen B.H. Kent

Reviewer #2 (Remarks to the Author):

The authors have addressed my comments and provided a new application for their protocol with the C-terminal modification of ubiquitin. I fully support publication in Nature Comm now.

Reviewer #3 (Remarks to the Author):

I have carefully read the revised manuscript by the N. Devaraj and coworkers, and the response letter to the referees, and I find the paper to be much better now. The revised version provided a detailed response to all comments, including citations to all suggested references. The addition work on Ub synthesis, together with Matt Pratt group was necessary to show the utility of this method for chemical protein synthesis.

In summary, I suggest the acceptance of this manuscript in its current state.

Congratulations.